# ToM2C: Target-oriented Multi-agent Communication and Cooperation with Theory of Mind

## Abstract

Being able to predict the mental states of others is a key factor to effective social interaction. It is also crucial to distributed multi-agent systems, where agents are required to communicate and cooperate with others. In this paper, we introduce such an important social-cognitive skill, *i.e.* Theory of Mind (ToM), to build socially intelligent agents who are able to communicate and cooperate effectively to accomplish challenging tasks. With ToM, each agent is able to infer the mental states and intentions of others according to its (local) observation. Based on the inferred states, the agents decide "when" and with "whom" to share their intentions. With the information observed, inferred, and received, the agents decide their sub-goals and reach a consensus among the team. In the end, the low-level executors independently take primitive actions according to the sub-goals. We demonstrate the idea in a typical target-oriented multi-agent task, namely multi-sensor target coverage problems. The experiments show that the proposed model not only outperforms the state-of-the-art methods in sample efficiency and target coverage rate but also has good generalization across different scales of the environment.

## 1 Introduction

Cooperation is a key component of human society, which enables people to divide labor and achieve common goals that no individual can reach on his/her own. In particular, human are able to form an ad-hoc team with partners and communicate cooperatively with one another [1]. Cognitive studies [2, 3, 4] show that the ability to model others' mental states (intentions, beliefs, and desires), called Theory of Mind (ToM) [5], is important for such social interaction. Considering a simple real-world scenario (Fig. 1), where three people (Alice, Bob, and Carol) are required to take the fruits (apple, orange, and pear) with shortest path. To achieve it, the individual will take four steps sequentially: 1) observing their surrounding; 2) Inferring the observation and intention of others; 3) communicate with others to share the local observation or intention if necessary; 4) making a decision and taking action to get the chosen fruits without conflict. In this process, the ToM is naturally adopted in inferring others (Step 2) and also guides the communication among agents (Step 3).

Motivated by this, machine learning researchers have takes efforts on developing the machine ToM [6] or modeling opponents [7] for multi-agent learning [8, 9, 10]. But most of the existing computing models are only used in toy environments, where are only a few agents (two or three) performing simple tasks. It is still challenging to implement such a thinking mechanism for social agents, especially in cases of many agents. That is because the mental state of one agent will be impacted by many other agents, leading to the accuracy and efficiency of the ToM drop.

In this paper, we study the Target-oriented Multi-Agent Cooperation problem (ToMAC). In ToMAC, the agents need to cooperatively reach and keep specific relations among the agents and targets. Such problem setting widely exists in real-world applications, *e.g.* collecting multiple objects (Fig. 1), navigating to multiple landmarks [11], monitoring a group of pedestrians [12]. When running,

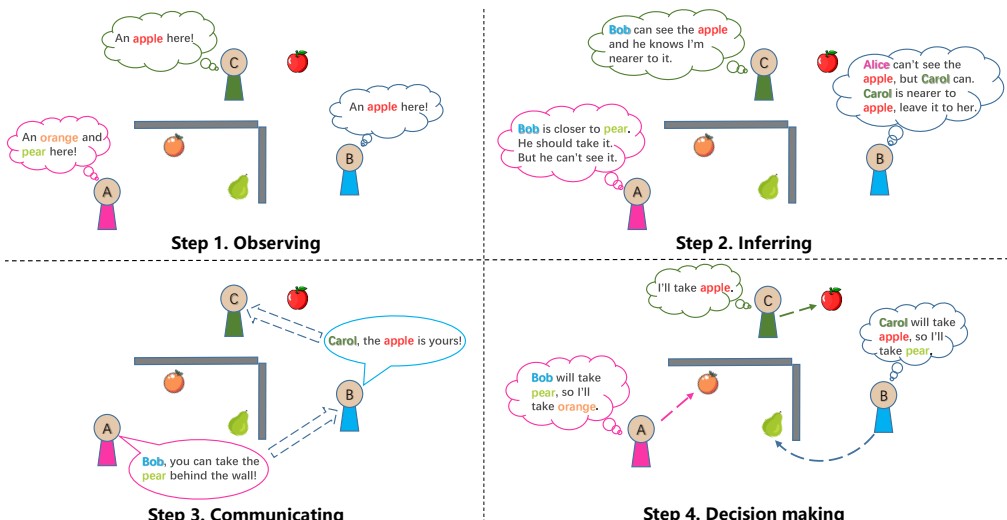

Figure 1: A fruits collection example. The agents are required to cooperatively collect the three target objects (apple, pear, and orange) in the room as fast as possible. The whole process can be divided into 4 steps. In the first step, 3 agents observes the environment and obtains the state of the visible targets. In the second step, each agent tries to infer what other agents have seen, and which targets they shall choose as goals. In the third step, each agent decides whom to communicate with according to the previous inference. In the fourth step, each agent decides its own goal of target based on what it observed, inferred, and received.

each agent is required to choose a subset of interesting targets and reaching them to contribute to the team goal. In this case, the key to realizing high-quality cooperation is **how to reach a consensus among agents** to avoid the inner conflict in the team. However, the existing multi-agent reinforcement learning methods still do not handle it well, as they only implicitly model others in the state representation and are inefficient in communication.

Here we propose a **Target-oriented Multi-agent Communication and Cooperation mechanism (ToM2C)** using the Theory of Mind, shown as Fig. 2. In ToM2C, each agent is of a two-level hierarchy. The high level policy (planner) needs to cooperatively choose certain interesting targets as a sub-goal to deal with, such as tracking certain moving objects or navigating to a specific landmark. Then low level policy (executor) takes primitive actions to reach the selected goals for $k$ steps. To be more specific, each agent receives local observation of targets, and estimate the local observation of others in the ToM Net. Combining the observed and inferred state, the ToM net will predict/infer the target choices (intentions) of other agents. After that, each agent decide 'whom' to communicate with according to local observation filtered by the inferred goals and the estimated observation of others. The message is rather simple and comprehensible, which is only the predicted goals of the message receiver, inferred by the sender. In the end, all the agents decide its own goals by leveraging the observed, inferred, and received information. Thanks to the inferring and sharing of intentions, the agents can easily reach a consensus to cooperatively adjust the target-agent relations to the expected.

Furthermore, we also introduce a communication reduction method to remove the redundant message passing among agents. Take the advantage of the centralized training decentralized execution (CTDE) paradigm, we measure the effect of the received messages on each agents, by comparing the output of the planner with and without messages. Hence, we can figure out the unnecessary connection among agents. Then we train the connection choice network to cut these dispensable channels in a supervised manner. Eventually, ToM2C systemically solves the problem of 'when', 'who' and 'what' in multi-agent communication, providing a compact, efficient and interpretable communication protocol.

The experiments are conducted in a challenging multi-sensor multi-target covering scenario. The team goal of sensors is to adjust their orientation to cover as many targets as possible. It is shown that our method achieves the highest coverage ratio among several state-of-the-art MARL methods [13, 12] in the case of 4 sensors and 5 targets. Moreover, we also demonstrate the strong scalability of ToM2C in different populations of sensors and targets. We further take an ablation study to evaluate the contribution of each key component of our model.

## 2   Related Work

**Multi-agent Cooperation.** The cooperation of multiple agents is crucial yet challenging in distributed systems. Agents' policies continue to shift during training, leading to non-stationary environment and difficulty in model convergence. Recent work [11, 14, 15, 16, 17] in multi agent reinforcement learning (MARL) mainly adopts centralized training decentralized execution (CTDE) paradigm to mitigate non-stationarity. However, such training method only implicitly guides agents to adapt to certain policy patterns of others. As a result, cooperation collapses even if there is only a slight change in the team formation, making the model extremely impractical and poor of scalability. Furthermore, some existing work tries to make use of communication to promote cooperation, such as [18, 19, 20]. Unfortunately, they all require a broadcast communication channel that pose a huge pressure on bandwidth. Besides, even though I2C [13] proposes a individual communication method, the message is just the encoding of observation, which is not only costly but also uninterpretable. Compared with existing methods, ToM2C does not only apply ToM to explicitly model intentions and mental states but also to improve the efficiency of communication to further promote cooperation. For the target-oriented multi-agent cooperation, HiT-MAC [12] propose a hierarchical multi-agent coordination framework to decomposes the target coverage problem into two-level tasks: assigning targets by centralized coordinator and tracking assigned targets by decentralized executors. The agents in ToM2C are also of a two-level hierarchy. Differently, thanks to the use of ToM, both levels are enabled to perform distributedly.

**Theory of Mind.** Theory of Mind is a long-studied conception in cognitive science [2, 3, 4]. However, how to apply the discover in cognitive science to building cooperative multi-agent systems still remains a challenge. Most previous work make use of Theory of Mind to interpret agent behaviours, but fail to take a step forward to enhance cooperation. For example, Machine Theory of Mind [6] proposes a meta-learning method to learn a ToMnet that predicts the behaviours or characteristics of a single agent. Besides, [21] studies how to apply Bayesian inference to understand the behaviours of a group and predict the group structure. [22] introduces the concept of Satisficing Theory of Mind, which means the sufficing and satisfying model of others. None of these work looks into the problem of multi-agent cooperation. [10] considers a 2-player scenario and employs Bayesian Theory of Mind to promote collaboration. Nevertheless, the task is too simple and it requires the model of other agents to do the inference. On the other hand, opponent modeling [7, 8, 9] is another kind of methods comparable with Theory of Mind. Agents endowed with opponent modeling can explicitly represent the model of others, and therefore plan with awareness of current status of others. Nevertheless, these methods rely on the access to the observation of others, which means they are not truly decentralized paradigms.

## 3   Methods

In this section, we will explain how to build a target-oriented social agent to realize efficient multi-agent communication and cooperation. We formulate the target-oriented cooperative task as a Dec-POMDP [23]. The aim of all agents is to maximize the team reward. Furthermore, agents are allowed to communicate with each other to enhance cooperation. The overall network architecture is shown in Fig. 2 from the perspective of agent $i$. The model is mainly composed of four functional networks: Observation encoder, ToM net, Communication choice net, and actor-critic net. To be specific, it receives a local partial observation $o_i$, which includes the information of visible targets. What's more, it obtains the current pose $(\phi_1, ..., \phi_n)$ of all the agents, where $n$ is the number of agents. The raw observation will be encoded into $E_i$ by an attention-based encoder. Then the agent starts to do Theory of Mind inference with the ToM net. It first estimates the observation representation $\epsilon$ of each other agents according to their poses. $\epsilon_j$ can be used for inferring the current visible targets of agent $j$, which is an auxiliary task that will be discussed later. Based on $\epsilon_j$ and $E_i$, agent $i$ infers the probability of agent $j$ choosing these targets as its goals, denoted as $G_{i,j}^*$. After that, agent $i$ decides whom to communicate with. We employ a graph neural network here as the communication choice net. The node feature of agent $j$ is the concatenation of $\epsilon_j$ and $E_i$ filtered by $G_{i,j}^*$. The final communication connection is sampled according to computed graph edge features. Agent $i$ will send $G_{i,j}^*$ to agent $j$ if there exists a communication edge from $i$ to $j$. Finally, $G_{i}^*, E_i$ and received messages is concatenated as $\eta_i$ for planner(actor) and critic. Planner $\pi_i^H(g_i|o_i)$ is the high level policy that chooses the goals $g_i$, which guides the low-level executor $\pi_i^L(a_i|o_i, g_i)$ to perform primitive actions.

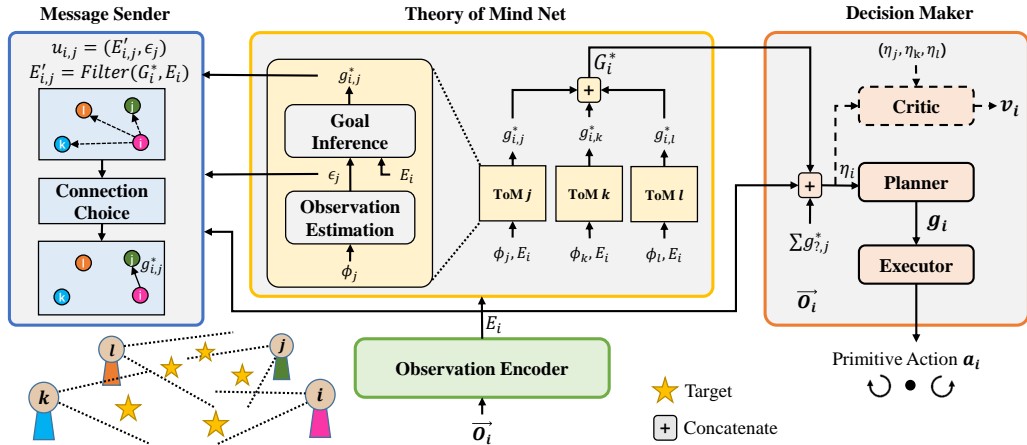

Figure 2: The architecture of ToM2C for each individual. There are 4 key components: Observation encoder, Theory of Mind net, Message sender and Decision maker.

In the following sections, we will illustrate the key components of ToM2C in details.

## 3.1 Observation Encoder

We employ the attention module [24] to encode the local observation. There are two prominent advantages of this module. On one hand, it is population-invariant and order-invariant, which is crucial for scalability. On the other hand, global information can be encoded into single feature due to the weighted sum mechanism. In this paper, we use scaled dot-product self-attention similar to [12]. $m$ is the number of targets. The input is the local observation $\vec{o}_i \in \mathbb{R}^{m \times d_{obs}}$ and the output is $\vec{E}_i \in \mathbb{R}^{m \times d_{att}}$, where $\vec{o}_{i,q}$ and $\vec{E}_{i,q}$ represent the raw and encoded feature of target $q$ to agent $i$ respectively.

## 3.2 Theory of Mind Network (ToM Net)

Inspired by the Machine Theory of Mind [6], we introduce ToM net that enables agents to infer the observation and intentions of others. Most previous work [7, 8, 10] consider two-player scenarios, where the agent only needs to model one other agent. Instead, we take a step forward to evaluate our model in a more complex multi-agent scenario consisting of n(>3) agents. Therefore, the entire ToM net of agent $i$ is actually composed of n-1 separate ToM nets, each utilized to model the corresponding agent. The single ToM net is made up of two functional modules: Observation Estimation and Goal Inference. The overall ToM net takes the poses of agents and local observation as input. Then it outputs the inferred observation representation and goals of others.

**Observation Estimation.** The first step of ToM inference is to estimate the observation representation of the other agent. The term refers to the visibility of the environment. Intuitively, when an agent tries to infer the intention of others, it should first infer which targets are seen by them. Take Bob in fig. 1 as an example. Before he tries to infer the goals of Alice and Carol, he first infers that Alice cannot observe the apple but Carol can. Similarly, Agent i infers the observation of agent $j$, denoted as $\epsilon_j$, with the pose $\phi_j$. Note that $\epsilon_j$ is only a representation of the observation. To better learn this representation, we introduce an auxiliary task here. Agent $i$ needs to infer which targets are in the observation field of agent $j$, based on this representation $\epsilon_j$ and local observation $\vec{E}_i$. In practice, we employ a GRU to model the observation of others on time series.

**Goal Inference.** After agent $i$ finishes the observation estimation of others, it is able to predict which targets will be chosen by them at this step. Just like human, the agent infers the intentions of others based on what it sees and what it thinks that others see. If we denote this goal inference network as a function GI, then the process can be formulated as : $G^*_{i,j,q} = GI(\vec{E}_{i,q}, \epsilon_j)$. $G^*_{i,j,q}$ stands for the probability of agent j choosing target q, inferred by i. Since there are a total of n agents and m targets in the environment, $\vec{G}^*_i \in \mathbb{R}_{(n-1) \times m}$.

With ToM net, each agent holds a belief on the observation and goal intentions of others. Such belief is not only taken into account for final self decision, but also serves as a indispensable component in communication choice. The details will be discussed in the next section.

## 3.3  Message Sender

Learning to communicate has been studied in a number of multi-agent reinforcement learning works. However, most of them either require a public communication channel or a centralized mechanism to decide the communication connection, which is definitely unrealistic for real multi-agent systems. Moreover, the message is usually an encoded feature, making it both uninterpretable and lengthy.

Instead, we introduce a message sender by leveraging the information inferred by ToM net. Each agent decides 'when' and with 'whom' to communicate completely on its own. And the message is the inferred ToM goals of the receiver. To achieve this, we use a graph neural network similar to [25, 26]. The details is in the next paragraph. After the model is trained, we further propose a communication reduction method to remove useless connections and improve the efficiency of the communication network.

**Inferred-goal Filter.** As stated before, we use Graph Neural Network (GNN) to learn the connection in an end-to-end manner. In previous works [27], there is only one global graph that collects all the observation as node features. Such implementation breaks the individuality. Instead, we propose a method to make use of the inferred state and intention to generate local graphs. Specifically, in the perspective of agent $i$, the feature of agent $j$ is the target features filtered by the inferred goals $G^*_{i,j}$ as follows. $\delta$ is a probability threshold, if $G^*_{i,j,q} > \delta$, then agent $i$ considers it as the goal that will be chosen by agent $j$.

$$E'_{i,j} = \sum_{q=1}^{m} (G^*_{i,j,q} > \delta) \cdot E_{i,q} \tag{1}$$

Then we concatenate the filtered feature $E'_{i,j}$ with the estimated observation representation $\epsilon_j$, to form the estimated node feature $u_{i,j} = (E'_{i,j}, \epsilon_j)$. For agent $i$ itself, $u_{i,i} = (\sum_q E_{i,q}, \epsilon_i)$, where $\epsilon_i$ is also computed by Observation Estimation module with the pose of $i$.

**Connection Choice.** For a scenario consisting of $n$ agents, there is a total of $n$ directed graphs $\mathcal{G} = (\mathcal{G}_1, \mathcal{G}_2, ... \mathcal{G}_n)$. $\mathcal{G}_i = (\mathcal{V}_i, \mathcal{E}_i)$ is the local graph for agent $i$ to compute the communication connection from agent $i$. The vertices $\mathcal{V}_i = \{f(u_{i,j})\}$, where $f$ is a node feature encoder. Edges $\mathcal{E}_i = \{\sigma(u_{i,j}, u_{i,k})\}$, where $\sigma$ is an edge feature encoder. Like the Interaction Networks (IN) [26], we propagate the node and edge features spatially to obtain node and edge effect. For convenience, we will describe only graph $\mathcal{G}_i$ in the following formula and omit the index $i$. Let $V_j$ be the encoded node feature of $j$, and $h_j$ be the node effect. Similarly, let $\mathcal{E}_{j,k}$ be the encoded edge feature, $h_{j,k}$ be the edge effect. Initially, $h_j = V_j, h_{j,k} = \mathcal{E}_{j,k}$. Then the graph iterates for several times to propagate the effect:

$$h_j = \Psi^{\text{node}}(V_j, h_j, \sum_k h_{k,j}) \tag{2}$$

$$h_{j,k} = \Psi^{\text{edge}}(h_j, h_k, h_{j,k}) \tag{3}$$

In the end, we obtain edge $(\mathcal{E}_{i,j}, h_{i,j})$, and compute the probabilistic distribution over the type of the edge (cut or retain). Here we apply the Gumbel-Softmax trick [28, 29] to sample the discrete edge type, so the gradients can be back-propagated in end-to-end training. Considering that it is the local communication graph of agent $i$, only the types of $\mathcal{E}_{i,-i}$ are sampled. If edge $\mathcal{E}_{i,j}$ is retained, agent $i$ will send a the inferred goals of $j$ to it.

**Communication Reduction (CR).** The communication choice network learns in an end-to-end manner. If no regularization is applied here, the network tends to learn a relatively dense communication connection graph. However, some of these connections are actually redundant. In fact, some receivers choose the same goals with and without these messages. Therefore, we can figure out the necessity of certain communication edges. Formally, we estimate the effect of the received messages to agent $i$ by measuring the KL-divergence between $g_i$ and $g_i^-$, referred as $\chi = D_{KL}(g_i^-||g_i)$. Note that $g_i^-$ denotes the probability distribution over the goals of agent $i$ when all the messages sent to $i$ are masked. If $\chi < \tau$, we regard that the messages are redundant to agent $i$. Thus the edges pointing at $i$ will be 'cut'. Otherwise ($\chi > \tau$), we 'retain' all the edges to agent $i$. Here $\tau$ is a constant, regarded

as a threshold. Supervised by the generated pseudo labels, the model learns to cut the redundant connections easily, leading to a more efficient communication network.

## 3.4 Decision Making

Once the agent receives all the messages, it can decide its own goals of targets based on its observation, inferred goals of others and received messages. Therefore, the actor-critic feature $\eta_i = (\vec{E}_i, \max_k \vec{G}^*_{i,k}, \sum_s \vec{G}^*_{s,k})$. The second term refers to the max inferred probability of an target to be chosen by another agent. The third term refers to the sum of the messages from others, indicating how much certain others infer that agent $i$ should choose the target. The actor decides its goals $g_i$ according to $\eta_i$. The centralized critic obtain global feature $(\eta_1, ... \eta_n)$ to compute value. The low level executor

$$\pi_i^L(a_i|o_i, g_i)$$

takes primitive action to accomplish the sub-goal. Although this executor can also be trained by reinforcement learning (RL) as [12], we find a simple rule-based policy can also work well in most cases. In this way, other methods without a hierarchical structure only need to learn the high-level policy, so we can compare them with our method fairly.

## 3.5 Training

The model can be divided as ToM net and other parts. ToM net is trained in supervised learning with the true state of others. Other parts are trained by reinforcement learning (RL). We adopt standard A2C [30] as the RL training algorithm, while any MARL method with CTDE framework is also applicable, such as PPO [31, 32].

**Learning ToM Net.** We introduce two classification tasks for learning the ToM Net, which is parameterized by $\theta^{\text{ToM}}$. First, the ToM net infers the goals $\vec{G}^*_i$ of others. Note that $g^*_{i,j,q}$ indicates the probability of agent $j$ choosing target $q$, inferred by $i$. Meanwhile, agent $j$ decides its real goals $g_j$. Therefore, $g_j$ can be the label of $g^*_{i,j}$. The Goal Inference loss is the binary cross entropy loss of this classification task:

$$L^{GI} = -\frac{1}{N} \sum_i \sum_{j \neq i} \sum_q [g_{j,q} \cdot \log(g^*_{i,j,q}) + (1 - g_{j,q}) \cdot \log(1 - g^*_{i,j,q})] \tag{4}$$

Secondly, the estimated observation representation $\epsilon$ is trained in the auxiliary task mentioned before. The agent $i$ infers which targets are in the observation of $j$, denoted as $c^*_{i,j}$. The ground truth is the real observation field $c_j$. $c_{j,q} = 1$ indicates that agent $j$ observes target $q$. Similar to the previous Goal Inference task, this Observation Estimation learning also adopts binary cross entropy loss:

$$L^{OE} = -\frac{1}{N} \sum_i \sum_{j \neq i} \sum_q [c_{j,q} \cdot \log(c^*_{i,j,q}) + (1 - c_{j,q}) \cdot \log(1 - c^*_{i,j,q})] \tag{5}$$

$$L(\theta^{\text{ToM}}) = L^{GI} + L^{OE} \tag{6}$$

**Training Strategy.** We find that it is hard for an agent to learn long-term planning from scratch. Therefore, we set the initialize episode length $L$ and discount factor $\gamma$ to a low value, forcing agents to learn short-term planning first. During training, the episode length and discount factor $\gamma$ increase gradually, leading the agents to estimate the value on a longer horizon.

Furthermore, we freeze the ToM net while the other parts of the model is updated through RL. The reason is that the ToM net infers the goals of others, and the policy network is continuously updated during RL training. Meanwhile, the output of ToM net is a part of the input to policy network. If we train them simultaneously, they are likely to influence each other in a nest loop. Therefore, we only collect the ToM inferred data into a batch during RL training. Once the batch is large enough, we stop RL and start ToM training to minimize ToM loss in Eq. 6.

## 4 Experiments

We evaluate ToM2C in the challenging multi-sensor target coverage problem. Sensors need to cooperate with others to reach a maximum target coverage rate. We compare our method with 3 state-of-the-art MARL methods: I2C [13], HiT-MAC [12], A2C [30], and a reference greedy search policy. We also conduct an ablation study to validate the contribution of ToM net and message sender. Finally, we show that our model can generalize to different size of agents and targets.

### 4.1 Environment

The environment is modified based on the one used in HiT-MAC [12], and it inherits most of the characters. As is shown in Fig.3, it is a 2D environment that simulates the real target coverage problem in directional sensor networks. Each sensor can only see the targets in the sector, if not blocked by any obstacle. There 2 types of target: destination-navigation and random walking. The former one moves in the shortest path to reach a previously sampled destination. The latter one moves randomly at each time step. At the beginning of each episode, the location of sensors, targets and obstacles are randomly sampled. Besides, the targets type is also sampled according to a pre-defined probability.

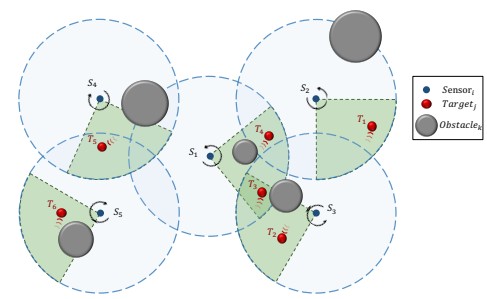

Figure 3: An example of the target coverage environment with obstacles.

**Observation Space.** At each time step, the local observation $o_i$ is a set of agent-target pairs: $(o_{i,1}, ...o_{i,m})$. If target $q$ is visible to agent $i$, then $o_{i,q} = (i, q, d_{i,q}, \alpha_{i,q})$, where $d_{i,q}$ is the distance and $\alpha_{i,q}$ is the relative angle. If target $q$ is not visible to $i$, then $o_{i,q} = (0, 0, 0, 0)$. Therefore, $o_i \in \mathbb{R}^{m \times 4}$.

**Action Space.** The primitive action for a sensor is to stay or rotate +5/-5 degrees. For our method, the high level action is the chosen goals $g_i$, which is a binary vector of length m. $g_{i,q} = 1$ means the agent chooses target $q$ as one of its goals. $g_{i,q} = 0$ means not. Although the low-level executor can be trained by reinforcement learning (RL) as [12], we find a simple rule-based policy can also work well in most cases. Therefore we only train the high-level policy. In this way, other methods without a hierarchical structure are comparable with our method.

**Reward.** Reward is the coverage rate of targets: $r = \frac{1}{m} \sum_q I_q$, where $I_q = 1$ if $q$ is covered by any sensor. If there is no target covered by sensors, we punish the team with a reward $r = -0.1$.

### 4.2 Baselines

We compare our methods with 4 baselines. HiT-MAC [12] is a hierarchical method that uses a coordinator to enhance cooperation. I2C [13] proposes a individual communication mechanism, which is also achieved by ToM2C. A2C [30] is a standard reinforcement learning algorithm. Here we employ A2C to train a single agent that selects the goals for all the sensors. Finally, we implement a heuristic search algorithm as a reference policy. This policy searches in one step for the primitive actions of all the sensors to minimize the sum of minimum angle distance of a target to a sensor.

### 4.3 Results

As fig.4(a) shows, ToM2C achieves the second highest reward (75) in the setting of 4 sensors and 5 targets, only lower than the searching policy (80). The vanilla A2C shows a similar performance to random policy, indicating that the task is not trivial. The reward performance of HiT-MAC is around 62, lower than the result presented in the original paper. This could be attributed to the addition of obstacles. I2C reaches a fair reward of 66, but we will show that such performance is still lower than our ablation models.

**Ablation Study.** We conduct this study to evaluate the 2 key components of our model: ToM net and Message sender. The ToM2C-Comm model abandons communication, so the actor makes decisions

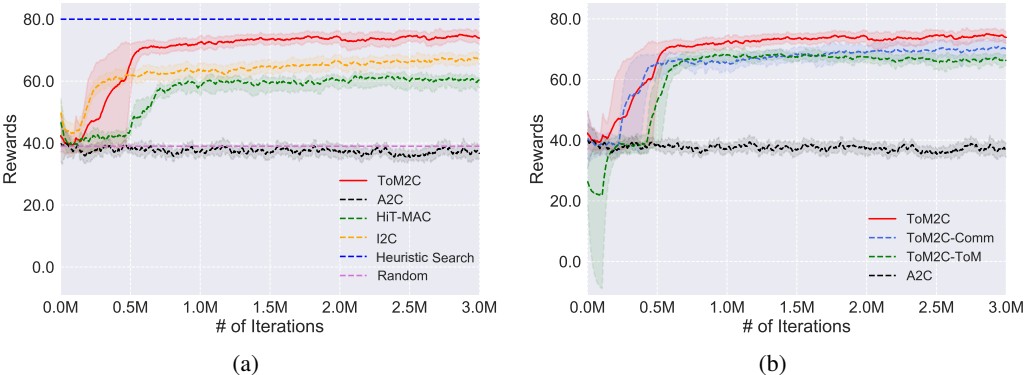

(a)                                                                    (b)

Figure 4: The learning curve of our method with baselines and reference policies. The learning-based methods are all trained in environment with 4 sensors and 5 targets. (a) comparing ours with baselines; (b) comparing ours with its ablations.

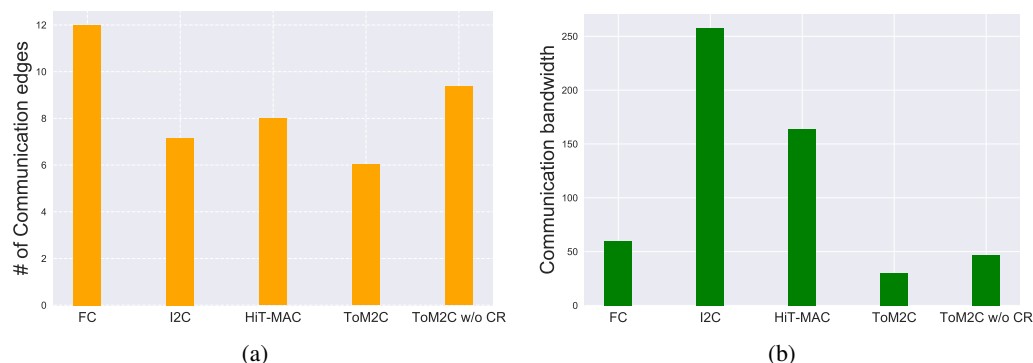

(a)                                                                    (b)

Figure 5: Communication performance analysis of our method compared with other algorithms. (a) comparison of the communication edges numbers; (b) comparison of the communication bandwidth.

only based on local observation and inferred goals of others. The ToM2C-ToM abandons ToM net, but keeps the Messages sender. However, as explained before, the local graph node feature is computed based on the ToM net output. To deal with this problem, we use the encoded observation $E_j$ to replace the original node feature $u_{i,j}$. In this way, the n local graphs degrades into one global graph, so the ToM2C-ToM model actually breaks the local communication mechanism. We show in fig.4(b) that if we abandon one of key components, the performance will drop. Specifically, ToM2C-Comm reaches 72, and ToM2C-ToM reaches 68, both higher than I2C. Considering that ToM2C-Comm outperforms ToM2C-ToM and ToM net is actually essential for communication, we argue that ToM net mainly contributes to our method.

**Communication Analysis.** We compare our method with several candidates in regard of communication expense. There are 2 metrics here: the number of communication edges and communication bandwidth. The latter metric considers both the count of edges and the length of a single message. There are 5 candidates here. FC refers to fully connected communication in ToM2C. ToM2C w/o CR refers to the ToM2C model without communication reduction. The communication in HiT-MAC is between the executors and the coordinator. As is shown in fig.5(a), ToM2C performs the least communication in regard of edge count, but this doesn't fully demonstrate the advantage of ToM2C over other methods. In fig.5(b), the communication bandwidth of ToM2C, ToM2C without CR, and even FC is much lower than I2C and HiT-MAC. It is because in ToM2C the message is only the inferred goals, while I2C and HiT-MAC have to send the local observation. Therefore, the single message in ToM2C is much simpler than that of I2C and HiT-MAC. As a result, the communication cost of ToM2C is extremely less than existing methods.

**Scalability.** We evaluate the scalability of our method to different number of sensors and targets. Note that the model is only trained in the setting of 4 sensors and 5 targets, so this could be regarded

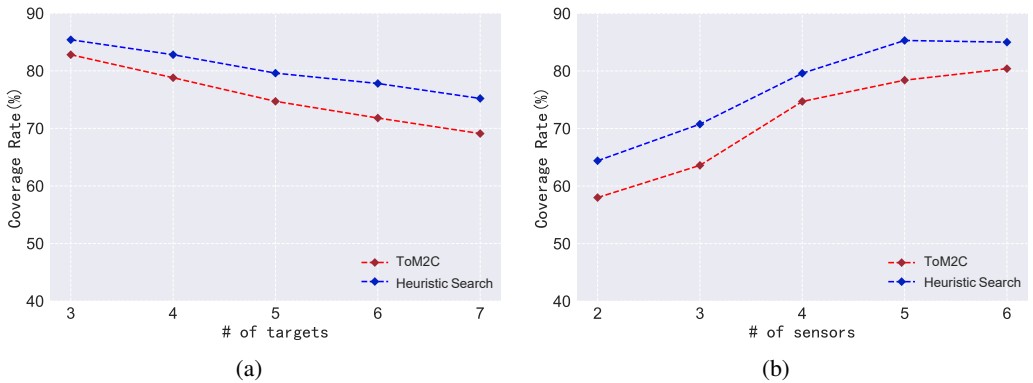

Figure 6: Analyzing the scalability of our method in scenarios with different sizes of sensors and targets. (a) $n=4$, $m$ is from 3 to 7; (b) $m=5$, $n$ is from 2 to 6

as zero-shot transfer. In fig.6(a), the number of sensors is fixed to 4, and in fig.6(b), the number of targets is fixed to 5. We also report the result of heuristic search because it is not learnt policy and has a good generalization in different settings. It is clear in the figures that the variation of coverage rate in ToM2C follows the trend of heuristic search when the difficulty of the setting changes. In this way, we show that ToM2C has rather stable generalization among different sizes of sensors and targets.

## 5 Conclusion and Discussion

In this work, we study the target-oriented multi-agent cooperation (ToMAC) problem. Inspired by the cognitive study in Theory of Mind (ToM), we propose an effective Target-orient Multi-agent Cooperation and Communication mechanism (ToM2C) for ToMAC. For each agent, ToM2C is composed of an observation encoder, ToM net, message sender, and decision-maker. The ToM net is designed for estimating the observation and inferring the goals (intentions) of others. It is also deeply used by the message sender and decision-making. Besides, an communication reduction method is proposed to further improve the efficiency of the communication. Empirical results demonstrated that our method can deal with challenging scenes and outperform the state-of-the-art MARL methods (I2C, HiT-MAC).

Although impressive improvements have achieved, there is still a number of limitations of this work leaving for addressed by future works. 1) The model is only evaluated in a simulated scenario. But the environment we used contains most features that other applications, e.g. partial observation, team reward structure. And each component in the model is general. So we are confident to extend ToM2C in other application scenarios, e.g. cooperative searching in the future. 2) Besides, the communication reeducation method can also be further optimized, as the pseudo labels we generated for communication reduction are noisy in some cases.

## Broader Impact

The target-oriented multi-agent cooperation problem widely exists in a lot of real-world applications. So a great number of robot tasks will benefit from our work, *e.g.* cooperatively searching for disaster victims, cleaning trash, scene reconstruction, actively capturing sports videos. They all will make our life more convenient and better. The use of ToM in multi-agent cooperation and communication will also promote the intersection of multi-agent systems and cognitive science, making them mutual benefit. But there is also the potential of being misused in the military field, *e.g.* using directional radars to monitor missiles/aircraft or controlling unmanned vehicles to attacks targets. If our method fails, some targets in the corner would be neglected by the agents.

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
