# OpenReview forum: "ToM2C: Target-oriented Multi-agent Communication and Cooperation with Theory of Mind"
_NeurIPS.cc/2021/Conference — NeurIPS 2021 Submitted_

### Official Review · Reviewer_koJ6 · 2021-07-16

**Rating:** 6
**Confidence:** 3

**Summary:**

The paper develops an approach to MARL that is inspired by theory of mind from psychology.
The authors claim that in problems like multi-sensor coverage that a better outcome (in terms of sensor coverage of
targets can be achieved) with such an explicit model that allows agents to communicate their "intentions".


**Ethical Concerns:**

- The authors finish their paper discussing potential misuse of the approach in military applications.
  This is not necessarily a misuse as military uses can be used to defend populations as well.
  Its worth thinking more broadly in terms of societal impact - these multi-agent UAV sensor coverage problems have many
  potential useful applications ranging from disaster management/rescue, to sports (think about motor sports or cycling
  or even many sports that will use UAV camers in the Olympic games), to law enforcement applications etc.

**Limitations And Societal Impact:**

- A more in depth discussion of the limitations of the approach would be desirable.
- I think the fruits collection example from Figure 1 actually detracts from the paper.
- In my opinion if the target coverage environment described in Section 4.1 was used
  from the start of the paper, it would be a more compelling and "real-world" example.
- The fruit example doesn't really add anything to the paper in my opinion.



**Main Review:**

# Originality
- The authors claim they introduce theory of mind (ToM) to multi-agent systems.
  There are a number of problems with this claim.

- First, the theory of mind is a theory from psychology that makes the assumptions that other
  have mental attitudes (such as beliefs, desires and intentions) such as our own, and by making
  this assumption we can better coordinate/communicate with other humans/agents.

- So the statement that the paper incorporates the theory of mind into a multi-agent system is not quite accurate
  and is perhaps misleading at worst.
  A more accurate statement would be that that the authors have incorporated ideas from the psychological theory of mind
  into a multi-agent system/architecture; or that they have developed a model of agent interaction that is **inspired** by
  the theory of mind from psychology. A such, the name of the model should be something else to distinguish it from the
  psychological theory.

- Second, there many other multi-agent systems (see the literature on multi-agent systems, BDI agents and multi-agent
  team work) that incorporate modelling other agent's beliefs, desires and intentions and in the case of team work
  modelling joint-beliefs, joint-goals and joint intentions.

- In any case the problem of having multiple sensors covering multiple targets is a well known problem that has had
  various approaches thrown at it. It has been addressed in both MARL and non-MARL (i.e. broader agent papers for a
  number of years). So it is isn't an original problem. Many agent systems have looked modelling the mental attitudes
  of team-mate agents. So, it isn't clear what the novel contribution is in this paper?
  Is it the combination MARL with ideas inspired by theory of mind? Explicitly stating this in the abstract/intro and
  conclusion would be a good idea.

- In a number of places you mention that agents share their "intentions". What is actually meant by this in this paper?
  Are agents sharing their policies?

- Note again that there is a body of work from the multi-agent systems community (specifically the field of BDI agents -
  that is agents based on the beliefs, desires and intentions model of agent reasoning) that have a very well defined
  meaning of intentionality. See works by Wooldridge, Roa/Georgeff on BDI agents for example.

## Quality
- Overall the paper is well structured, the paper reads well and there are informative diagrams.
- As mentioned above, the omission of broader references from work in multi-agent systems community on modelling shared
  mental attitudes is an issue but otherwise the related work section seems ok.
- The method section seems ok. However, it would seem to me that the proposed solution seems overly complex for the
  examples used in the experiments. I have a suspicion that a simpler approach could have garnered similar results for
  the simple domains shown. What would be of more interest would be a domain where it is obvious that a could outcome
  could not be achieved without explicit cooperation/communication.
- So a possible critique would be, could the problems be solved witout a ToM-Net; or why do you need a neural network
  implementation that is inspired by ToM?
- The experiments section is good. The explanation of the environment is good, with explicit descriptions of the
  reward, action spaces and observation space.
- While the baseline comparison is good, I think a better comparison would be your own architecture with and without a
  model inspired by theory of mind.
- Should there be error bars on the bar charts in Figure 5?
- Also good to see the results shown in Figure 6 where the number of targets and the number of sensors is increased.
  A couple of questions here:
  1) It seems the heuristic search outperforms the ToM2C model; I was wondering if this was the case (as per my previous
  comment) have you over engineered the architecture where a simpler model achieves a better result. What is the
  advantage or contribution of your approach then?
  2) Should there be error bars in Figure 6 (how many cases were run)?
  3) The numbers of targets and sensors is still within the same order of magnitude. A more impressive scalability
     metric would be if you scaled the number of sensors and targets by orders of magnitude (e.g. 10, 100, 1000 etc)
     (or perhaps 5, 10, 50, 100, 500, 1000). You could then plot number of sensors vs number of targets as a 2D plot
     and have the % coverage rate as a heatmap. That would be interesting to see.
- It was good to see Section 5 that includes a Conclusion and Discussion and even a broader impact section.
  My recommendation for these sections would be to reorganise them slightly. Rename Section 5 Discussion and
  rename "Broader Impact" to Conclusions and cover the following:
- 5. Discussion (discussion as is).
- 6. Conclusion (include a small discussion on limitations, future work and broader impact).


## Clarity
- There are some parts of the paper where sentences read somewhat awkwardly, but nothing major in my opinion.
- I found the paper fairly clearly to read, but like all papers can also benefit through another edit just to make sure
  everything is clear for the reader.

## Significance
- Multi-sensor target coverage is well studied program across many different AI literatures.
- I'm not sure the results here are significantly better than heuristic search or some of the other baselines
  to conclusively state that the approach taken will make a significant impact on the field.
- Note, a variation on my previous critique - this isn't to say that incorporating models inspired by the theory of mind is a
  bad idea in multi-agent systems. On the contrary, I and many others believe that it is a good idea.
  However, a neural network might not be the best way to do it.


**Time Spent Reviewing:**

2

---

> ### Author Response · Authors · 2021-08-10
> **Response to Reviewer koJ6**
>
> Thanks for your valuable comments. The following is our response to your concerns.
> - **Q1: The statement that the paper incorporates the theory of mind into a multi-agent system is not quite accurate.**
> A1: Theory of mind *"refers to humans’ ability to represent the mental states of others, including their desires, beliefs, and intentions." (machine ToM[6])*. So we argue that ToM is actually not a "theory" but a human-like ability. What we wanted to state is that we realized a learnable model that endows agents the ability to infer the observations and intentions of others. And the multi-agent cooperation and communication are further beneficial from such machine ToM. As you mentioned, our ToM-net is indeed inspired by the concept from psychology. So we will consider modifying our statement and the name of the model in the future edition. Thank you for the suggestion!
>
> - **Q2: What is the advantage or contribution of your approach?**
> A2: The contribution of this paper can be summarized in three-fold:
> 1) We introduced a ToM-inspired model for scalable target-oriented multi-agent cooperation. Specifically, the individual agent is of a two-level hierarchy. The ToM net at the high-level policy focuses on estimating the intention (which targets will be chosen) and observation (which targets are observed) of others. Thanks to the attention mechanism, it can easily handle the variable input/output size and extend to large-scale multi-agent systems (n>3). Note that previous works in this scope (Bayesian Theory of Mind for example) are basically limited to the size of 2 or 3 agents because the action/inference space will grow exponentially as the size of agents grows, leading to a fatal failure.
> 2) We provided a ToM-based communication mechanism, which is fully decentralized in execution. We also proposed a communication reduction method to further improve communication efficiency. To the best of our knowledge, no previous ToM-related literature discussed how to leverage ToM to provide the communication protocol for ’when’, ’who’ and ’what’ in multi-agent communication. Although I2C also introduced an individual communication mechanism, it is not based on ToM and is much less efficient compared with ToM2C (see Fig. 5).
> 3) We conducted several experiments in a challenging multi-agent target coverage task and showed that our model not only outperforms the state-of-the-art MARL methods in coverage rate and communication efficiency but also shows good scalability across scenes of different populations. We also proposed two training strategies to benefit learning.
> Thank you for the suggestion to explicitly state our contribution in the paper. We will add it in our next edition.
>
> - **Q3: Could the problems be solved without a ToM-Net; or why do you need a neural network implementation that is inspired by ToM?**
> A3: In the original submission, we already reported the ablation study in **Sec 4.3**, the comparison is shown in **Fig.4(b)**. The results show that both ToM net and the Message sender contribute to the improvement of multi-agent cooperation. The neural network implementation is necessary for learning a generalizable state representation in complex environments, where are multiple obstacles and moving targets.
>
> - **Q4: About intention, what is actually meant by this in this paper?**
> A4: As we stated in the introduction, L 50, "intentions" refer to the target choices (sub-goals) of agents. For example, if agent $i$ infers that agent $j$ will choose target $p$ and target $q$ as its goals, and agent $i$ decides to communicate with agent $j$, then agent $i$ will send the inferred intentions of agent $j$ to it. Therefore, it may be more accurate to change the term into "share the inferred intentions (the output of the ToM net)". We will clarify this in the next version.
>
> - **Q5: Should there be error bars on the bar charts in Figure 5? Should there be error bars in Figure 6 (how many cases were run)?**
> A5: Thanks for your suggestion. We will add the error bars in the revision. For each setting, we run with 3 different seeds to report the results.
>
> - **Q6：The proposed solution seems overly complex for the examples used in the experiments.**
> A6: The multi-target tracking problem is not easy to solve. The environment is partially observable and is dynamic due to the motion of targets, so trivial methods like A2C will fail (see Fig. 4). The reason why heuristic search outperforms the other methods is that it exploits the global state to do the global search, while the agents in all the RL-based methods only have access to local observations. Therefore, simpler approaches without communication or certain cooperation mechanism are not likely to solve this problem.
>
> - **Q7: ...plot number of sensors vs number of targets as a 2D plot and have the % coverage rate as a heatmap.**
> A7: Thank you for the suggestion! We will consider testing our model on a larger scale and plot the heatmap in our next edition.

---

> > ### Comment · Reviewer_koJ6 · 2021-08-19
> > **Reply to authors comments**
> >
> > Thank you for taking time to reply to my suggestions from my review.
> > I appreciate I've somewhat laboured the point regarding the claims about theory of mind and making the distinction between theory of mind in psychology and building an agent model that takes inspiration from this theory.
> > Similarly for your definition of intention. There are well defined models of intention in the multi-agent systems community.
> > For example in the BDI model of agent reasoning, intentions are a way of representing committing to a specific plan to achieve a goal.
> > Regarding your reply to Q4: are you really sharing "inferred" intentions or are you sharing goals between agents?

---

> > > ### Author Response · Authors · 2021-08-20
> > > **Reply to Reviewer koJ6**
> > >
> > > Thank you for taking time to reply to our response. Referring to the definition in the BDI model,  intentions represent the deliberative state of the agent – **what the agent has chosen to do**. In the multi-target multi-sensor coverage problem, the goal represents **"what the agent has chosen to track"**. Note that agents act hierarchically to perform tasks. To be more specific, the planner (high-level policy) chooses some targets as its goals for the next k steps, and the executor(low-level policy) takes primitive actions for the next k steps based on the chosen targets (goals).
> > >
> > > Similarly, in the cooperative navigation problem, the goal indicates **"what the agent has chosen to reach"**. Therefore, we argue that *the goal proposed by the planner is an abstraction of the real intentions of the agent in the target-oriented problems*. So for your question, *ToM2C agents really share the "inferred intentions" with others*, which are the "inferred goals" of other agents.

---

> ### Author Response · Authors · 2021-08-30
> **Note to Reviewer koJ6**
>
> In our response, we provided the details requested about the method and answered the questions. Please let us know if any of your concerns were not addressed by our response.

---

> > ### Comment · Reviewer_koJ6 · 2021-09-01
> > **Update review score**
> >
> > Hello authors,
> > Apologies for the late reply. I have gone through your comments again and I have adjusted my review score from a 5 to a 6.

---

### Official Review · Reviewer_2f7u · 2021-07-16

**Rating:** 5
**Confidence:** 4

**Summary:**

This paper proposes a multi-agent communication and cooperation framework, ToM2C, where agents need to decide who and when to communicate with, and how to allocate subtasks (specifically, subtasks are defined by physical locations that agents need to navigate to / cover). The key idea is to train the agents to infer others' observations and goals, which may in turn helps achieve more effective and efficient communication/cooperation. The method adopts ToMnet-like architecture for a ToM module, and uses GNNs to fuse information and inference from multiple channels. Compared to the previous GNN architectures for multi-agent policies, it proposes two new designs, inferred-goal filter and communication reduction, which are enabled by the goal inference. The evaluation was done in a 2D continuous environment simulating a target coverage task. Compared to recent approaches, HiT-MAC and I2C, the proposed method reaches a higher reward with slightly fewer messages in this single task.

**Limitations And Societal Impact:**

I would like to see more discussions on the limitations of the proposed method and how it is grounded in prior literature.

**Main Review:**

Strengths:
+ The work is well motivated. Indeed, ToM is a key ability for human cooperation and communication, which may also help machine agents' interactions.

+ The method shows improvement over 2 recent baselines, in terms of rewards and communication costs, which is encouraging.

Weaknesses:
- It is unclear to me how ToM2C as a problem setup differs from TarMAC [19] and I2C [13]. If it is conceptually the same setup, where an agent will send messages directed to individual agents, then I do not see how this is a different framework.

- Training ToMnet module requires other agents' ground-truth observations (what entities they can see) and goals.

- This is not the first paper that introduces ToM (goal inference, belief inference) to multi-agent RL, e.g., the suggested references [1,2,3].

- Many design choices such as how goal is represented are restricted to target navigation / coverage type of tasks. Generalizability seems quite limited compared to existing methods (e.g., I2C). This also raises the question -- how much benefit we see in the experiment comes from these domain knowledge (decomposing the task into specific subtasks defined by the target locations) and additional GT supervision that are not available in existing methods. The proposed method does seem to enjoy an unfair leg up when considering existing methods need to train a policy from scratch without knowing what subtasks are.

- Related to the above point, what if we do not have manually defined subtasks?

- Given manually defined subtasks, the suggested reference [4] proposed a Bayesian approach for ToM-based cooperation, which is related and should be discussed in the paper.

- The evaluation was done in only one experiment. It is fine if the scope of the work is to solve the target coverage problem, but this intended scope should be clarified. Otherwise, it is unclear how this method could be applied to other tasks and environments, since there are domain-specific knowledge and supervision used for model design and training.

- I do not see the number of runs conducted to produce the results. I would also expect to see variance in Figure 5 & 6.

Minor comments:
- fig.x -> Fig. x

Suggested references:

[1] Mordatch & Abbeel, Emergence of Grounded Compositional Language in Multi-Agent Populations, AAAI 2018.

[2] Qi & Zhu, Intent-Aware Multi-Agent Reinforcement Learning, ICRA 2018.

[3] Shu & Tian, M^3RL: Mind-aware Multi-agent Management Reinforcement Learning. ICLR 2019.

[4] Wu et al., Too many cooks: Bayesian inference for coordinating multi-agent collaboration. CogSci 2020.

Post Rebuttal:
I would raise my rating to 5. I think a major revision is necessary so the submission is not ready in its current form.

**Time Spent Reviewing:**

3.5 hours

---

> ### Author Response · Authors · 2021-08-10
> **Response to Reviewer 2f7u**
>
> Thanks for your valuable comments. The following is our response to your concerns.
>
> - **Q1: How ToM2C as a problem setup differs from TarMAC [19] and I2C [13].**
> A1: In the view of multi-agent communication, TarMAC is different from I2C, and ToM2C is closer to I2C. As mentioned by the authors from I2C, *"TarMAC is still a traditional broadcast approach (**all-to-all communication**) with the attention that allows agents to turn a blind eye to received inconsequential messages."* Instead, I2C realizes the **real individually controlled communication**, where each agent independently decides whom to communicate with. Thus, our setting is similar to I2C. Differently, our agent is of a two-level hierarchy and makes use of ToM Net to provide the communication protocol for ’when’, ’who’ and ’what’ in multi-agent communication, while I2C only focuses on 'who' and is not ToM-related. As a result, ToM2C significantly outperforms I2C in the cost of communication bandwidth.
>
> - **Q2: Training ToMnet module requires other agents' ground-truth observations (what entities they can see) and goals.**
> A2: The **centralized training and decentralized execution** training strategy is widely leveraged in most popular MARL methods (MADDPG, QMIX, ...). The ground truth observations and goals are only used for training the ToM net. Therefore, requiring other agents' observation and goals in training is not a weakness compared with most MARL algorithms. On the contrary, it should be our strength as appreciated by Reviewer RvLm, *"The communication reduction method used by the agents in the team that uses centralized training decentralized execution is interesting and makes the contributions novel."*
>
> - **Q3: This is not the first paper that introduces ToM (goal inference, belief inference) to multi-agent RL.**
> A3: First, to the best of our knowledge, no previous ToM-related literature (including [1][2][3]) discussed **how to leverage ToM to guide decentralized communication and improve communication efficiency for multi-agent cooperation**. Secondly, previous works in this scope are basically limited to the size of 2 or 3 agents, while ToM2C can be easily extended to a rather large scale (n>3). In this respect, our method is quite different from previous work and the multi-agent community will benefit from ToM2C.
>
> - **Q4: Generalizability seems quite limited compared to existing methods.**
> A4: The target-oriented multi-agent cooperation problems we focus on have a wide range of variety in real-world applications (also acknowledged by reviewer koJ6). Most environments with multiple target objects and agents are all in this scope, e.g., cooperative navigation, hunting, and disaster rescue. We choose the multi-sensor multi-target coverage problem in our experiment because it is both challenging and representative. To better show the generalization of our method in other tasks, we applied our method with minor revision in the 7v7 Cooperative Navigation environment (Lowe et al. '17) and achieved a competitive episode-average reward (-0.796 ± 0.026) against MADDPG (-2.748 ± 0.610) and I2C (-1.597 ± 0.402). It is also an interesting future direction to apply ToM2C to other multi-target multi-agent cooperation problems.
>
> - **Q5: How much benefit comes from these domain knowledge (decomposing the task into specific subtasks defined by the target locations).**
> A5: We mainly focus on target-oriented multi-agent cooperation problems in this paper, which naturally contains the division of targets. As we mentioned in Sec. 4.1, line 267, "we only trained the high-level goal selector, and the low-level policy is rule-based." It should be clarified that all the other methods, including HiT-MAC, I2C and A2C, also benefit from the hierarchy structure and share the low-level rule-based policy. For example, the I2C policy only decides which targets to choose as goals, and this high-level action will be sent to the shared low-level rule-based policy to generate low-level action. Therefore, **all the methods share the benefit of targets division**, which means it is not an unfair comparison. Besides, you can also refer to the experiments in HiT-MAC to see that how weak the existing MARL methods are when running without the hierarchy, even there is no obstacle in the environment used in HiT-MAC. So we argue that such a definition is reasonable and valuable for the target-oriented MARL, and most MARL methods will benefit from it.
>
> - **Q6: The suggested reference [4] proposed a Bayesian approach for ToM-based cooperation, which is related and should be discussed in the paper.**
> A6: The Bayesian Theory of Mind approach you mentioned indeed enhances multi-agent cooperation. However, it assumes that an agent can only select one sub-task at the same time, while ToM2C agent is not restricted to selecting only one target. Furthermore, the mentioned paper focuses on the environments that "have a partially ordered set of sub-tasks", while our target coverage environment does not have such property. As a result, this method is unable to implement in the target coverage environment to compare with ToM2C.
>
> - **Q7: The number of runs conducted to produce the results. Variance in Figure 5 & 6.**
> A7: For each setting, we run with 3 different seeds to report the results. We will add the variance in the revision.

---

> > ### Comment · Reviewer_2f7u · 2021-09-01
> > **Thanks for your responses**
> >
> > Thanks for your responses and additional experimental results! I now understand the experimental setup and the benefit provided by the approach better, and will slightly raise my rating. The reason why I am still leaning towards rejection at this time is that the missing discussions on prior work, and the additional experimental results are very important for people to really understand how this work is different from the prior work, what were the settings for the experiments, and how general this approach is (i.e., what kinds of problems could benefit from it). There should be a major revision to address these points, both in literature review and in experiments. The current submission is not ready in my opinion.
> >
> > Minor point:
> > For Q3, it is actually not true that prior work is only limited to 2 to 3 agents. There are works that enable more agents, such as [3] in the suggested references.

---

> > > ### Author Response · Authors · 2021-09-02
> > > **Reply to Reviewer 2f7u**
> > >
> > > Thanks for your constructive comment! We will enrich the discussions on prior work, clarify the details and supplement the additional experimental results in the revision.
> > >
> > > For the minor point you mentioned, the main difference between the referenced paper [3] and ours is the organize mechanism (**centralized vs. decentralized**). [3] proposed to train a centralized agent(manager), which needs to collect all the observations of workers, and assign sub-tasks to workers for multi-agent coordination.  Therefore, this method should be regarded as a centralized manner, in which *real-time high-bandwidth communication between the manager and workers is required*. Such a framework is impractical in most real-world scenarios. Differently, we focus on a **fully decentralized manner**, where agents have to make decisions independently with their own local observation and limited message from others. It is more practical but also challenging. However, to the best of our knowledge, we do not find any prior works on Theory of Mind that can well address the distributed multi-agent communication and cooperation on a larger scale(n>3).
> > > So it is unfair to compare the population of agents with [3]. We will clarify this in the revision.

---

> ### Author Response · Authors · 2021-08-30
> **Note to Reviewer 2f7u**
>
> In our response, we provided the details requested about the method, updated evaluation results and answered the questions. Please let us know if any of your concerns were not addressed by our response.

---

### Official Review · Reviewer_ADoL · 2021-07-19

**Rating:** 5
**Confidence:** 3

**Summary:**

This paper addresses the problem of reasoning about the mental states of other agents in multiagent reinforcement learning using Theory of Mind, focused on cooperative MAS domains such as multi-sensor target tracking.  A deep learning architecture is proposed that combines four networks to create a centralized solution: (1) one network that uses attention to encode observations for communication messages, (2) another than estimates the observations and goals of the agents in the environment (generalizing prior work to more than two cooperating agents), (3) a third that learns a graph determining who should communicate with whom, and when communication should happen, and (4) an actor-critic network for determining the policies of agents.  Experimental results varying the number of sensors and targets in the environment demonstrate the performance of the proposed solution against baselines.

**Limitations And Societal Impact:**

The broader impacts were discussed in the paper, although also addressing limitations of the research would strengthen the overall work.

**Main Review:**

===Originality===

The problems of multi-sensor target tracking in particular, and cooperative multiagent reinforcement learning are popular, important problems in the MAS and reinforcement learning communities.  The solution is an incremental combination of different types of neural networks to produce a new solution that has some differences from existing work.  I didn't fully understand the novelty of the solution as I was reading the paper.  The generalization to environments with more than two agents is important (albeit straightforward), and prior work seems to focus individually individually on the networks that make up the components of the proposed solution, so I think the proposed solution is more holistic than prior work, but I am not certain.

===Quality===

The related work seemed mostly complete, although I was left wondering why ToMnet wasn't chosen as a baseline, given the similarities with the focus of this work.  I also wasn't sure why A2C was chosen as the underlying algorithm and baseline, when PPO is typically used instead.

Another question I had about the model while reading was in Eq. 1, why is \delta used?  Wouldn't argmax G^* make more sense?  What happens if 2+ goals have G^* higher than \delta?  What happens if 0 goals have G^* higher than delta?

===Clarity===

Overall, the paper was relatively easy to follow, and I think I understood most of the details (I do have some questions below in the Significance section of this review).  Some clarifications would be helpful, such as more explicitly explaining the novelty of the solution -- what is unique vs. what is simply a combination of known techniques?  Also, a running illustrative example in Section 3 would help the reader understand what is the terms refer to and what is being learned.  Since you only have one test domain, it would be a great choice here.

===Significance===

My greatest uncertainty after reading the paper is about the significance of its contributions to the literature.  As mentioned above, I'm not exactly sure what the novelty in the work is.  Is it just a new holistic deep architecture?  Are some of the parts meaningful advances over the state-of-the-art beyond considering more than 2 agents?  To me, it appears to be a combination of known techniques with some incremental advances.

Also, only one domain was considered, so it is difficult to assess how the benefits of the approach might generalize.  Is it only useful in target-tracking domains?  Could it be applied to other cooperative MAS, which are a hot topic in deep MARL right now?  This is especially complicated by the fact that the domain appears to be relatively simple to solve since (1) simple rule-based policies were appropriate instead of needing reinforcement learning to learn low-level policies, and (2) a heuristic search algorithm outperformed all of the reinforcement learning solutions.  In real-world applications, we choose RL-based solutions when learning is necessary, possibly because the system needs to adapt to the dynamic environment, or when insufficient information is available a priori to know how to operate.  In this case, RL wasn't necessary, so it's difficult to know if the proposed approach outperformed the others because it achieved superior learning, or if it exploited some of the same reasons that heuristic search did well without learning.  So I'm left wondering if I'm misunderstanding something -- did heuristic search have access to information that wasn't available to the RL methods so that it is an appropriate upper bound on performance?  More details would help clarify here, which changes how the significance of the work might be viewed by the reader.

I was also unsure exactly how the scalability experiments were conducted.  You state that "Note that the model is only trained in the setting of 4 sensors and 5 targets, so this could be regarded as a zero-shot transfer".  How then did you conduct the experiments whose results are presented in Figure 6b?  If you have a different number of sensors than 4, wouldn't you need a different Theory of Mind Network, since it has components for each of the n agents in the environment?  Wouldn't this also affect the other components of the network since the communication graphs would be different?

**Time Spent Reviewing:**

4

---

> ### Author Response · Authors · 2021-08-10
> **Response to Reviewer ADoL**
>
> Thanks for your valuable comments. The following is our response to your concerns.
>
> - **Q1: The novelty/contribution is unclear.**
> A1: The contribution of this paper can be summarized in three-fold:
> 1) We introduced a ToM-inspired model for scalable target-oriented multi-agent cooperation. Specifically, the individual agent is of a two-level hierarchy. The ToM net at the high-level policy focuses on estimating the intention (which targets will be chosen) and observation (which targets are observed) of others. Thanks to the attention mechanism, it can easily handle the variable input/output size and extend to large-scale multi-agent systems (n>3). Note that previous works in this scope (Bayesian Theory of Mind for example) are basically limited to the size of 2 or 3 agents because the action/inference space will grow exponentially as the size of agents grows, leading to a fatal failure.
> 2) We provided a ToM-based communication mechanism, which is fully decentralized in execution. We also proposed a communication reduction method to further improve communication efficiency. To the best of our knowledge, no previous ToM-related literature discussed how to leverage ToM to provide the communication protocol for ’when’, ’who’ and ’what’ in multi-agent communication. Although I2C also introduced an individual communication mechanism, it is not based on ToM and is much less efficient compared with ToM2C (see Fig. 5).
> 3) We conducted several experiments in a challenging multi-agent target coverage task and showed that our model not only outperforms the state-of-the-art MARL methods in coverage rate and communication efficiency but also shows good scalability across scenes of different populations. We also proposed two training strategies to benefit learning.
>
>
> - **Q2: Why ToMnet wasn't chosen as a baseline?**
> A2: The ToMnet is only introduced for predicting the trajectory/behaviour of agents, instead of the multi-agent cooperation. It is actually used for modeling other agents and doesn't further make use of such inference to guide self action and communication. So it is not an easy thing to directly implement the ToMnet on the target-oriented multi-agent cooperation problem for comparision. In fact, the ablation of our method (ToM2C-Comm, which models others but without communcation in cooperation) can be viewed as a variant of the ToM-like net for multi-agent cooperation.
>
> - **Q3: Why A2C was chosen as the underlying algorithm and baseline?**
> A3: As we mentioned in our paper in section 3.5, line 217, "any MARL method with CTDE framework is also applicable, such as PPO". In fact, we did try PPO as the underlying algorithm and the result showed that there's nearly no difference between PPO and A2C.
>
> - **Q4: In Eq.1, why is $\delta$ used? Wouldn't argmax $G^\*$ make more sense? What happens if 2+ goals have $G^\*$ higher than $\delta$? What happens if 0 goals have $G^\*$ higher than delta?**
> A4: Intuitively, when agent $i$ decides whether to communicate with agent $j$, it should consider all the targets that agent $j$ may choose as goals. In the cooperative navigation problem, it is reasonable to use argmax, as each agent only needs to choose one goal to reach. But in the target coverage problem, it is possible that an agent chooses multiple targets, so argmax $G^*$ is not proper. We set the probability threshold $\delta$ to $0.5$, which means if agent $i$ infers that target $q$ has a probability of more than $0.5$ to be chosen by agent $j$ as its goal, the feature of target $q$ will be added to the node feature of agent $j$ in the local graph $i$. If 2+ goals have $G^*$ higher than $\delta$, then the features of these goals will be summed up. If no goals have $G^*$ higher than $\delta$, then the sum should be a zero vector.
>
> - **Q5: Is it only useful in target-tracking domains? Could it be applied to other cooperative MAS, which are a hot topic in deep MARL right now?**
> A5: ToM2C can be applied to other target-oriented cooperative multi-agent tasks, such as Cooperative Navigation, Predator&Prey (2 teams) and other real-world applications like disaster rescue. To be more convincing, we applied our method in the 7v7 Cooperative Navigation environment (Lowe et al. '17) and achieved a competitive episode-average reward (-0.796 +- 0.026) against MADDPG (-2.748 +- 0.610) and I2C (-1.597 +- 0.402). It is an interesting future direction to apply ToM2C to other multi-target multi-agent coordination problems.
>
> - **Q6: The domain appears to be relatively simple to solve.**
> A6: The multi-target coverage problem is not a trivial problem. There are multiple moving targets (highly dynamic environment), the observable range of the sensor is limited (partial observation), and they are distributed (decentralized control). So it is challenging and MARL is needed. Note that previous works mainly study this problem in a relatively simple setting, where the agent can get the global state of the environment or the targets are static. The heuristic search method used in our experiment runs in such a simple setting, allowed to access the global state in each step. We regard it as a reference upper bound rather than a baseline to beat. Besides, you can also refer to the experiments in HiT-MAC to see how weak the existing non-hierarchical MARL methods are in this domain, even when there is no obstacle in the environment used in HiT-MAC.
>
> - **Q7: How were the scalability experiments conducted?**
> A7: Although the ToM network of an agent has ToM models for each of the other agents, these models are actually shared because all the agents are homogeneous. Therefore, when we change the number of agents to test the scalability, we simply alter the number of ToM models in the ToM network. As for the communication graphs, they are locally computed and the propagating mechanism is immuned from varying number of nodes. Besides, we employ an attention module as the observation encoder, which is population-invariant. Based on all the mechanisms described above, we can simply transfer the learned model to different scales.

---

> > ### Comment · Reviewer_ADoL · 2021-08-23
> > **Response to Authors**
> >
> > I thank the authors for their thorough responses to my questions and points of confusion.  I think I understand the contributions and significance of the paper better now.
> >
> > From A3: _In fact, we did try PPO as the underlying algorithm and the result showed that there's nearly no difference between PPO and A2C._
> >
> > Was PPO also attempted as the baseline?  PPO, being a newer algorithm, sometimes outperforms A2C, which could affect how your approach does against other baselines.  If you attempted PPO as a baseline, those results could be worth including since it shows that you tried the best possible baselines, increasing the significance of your approach's results.
> >
> > From A5: _To be more convincing, we applied our method in the 7v7 Cooperative Navigation environment (Lowe et al. '17) and achieved a competitive episode-average reward (-0.796 +- 0.026) against MADDPG (-2.748 +- 0.610) and I2C (-1.597 +- 0.402)_
> >
> > If you can add these results to an appendix, it will make your results more convincing since (1) you used a second domain in a different type of environment from another study (demonstrating generalizability of your approach), and (2) you outperformed useful baselines.
> >
> > From A6: _The heuristic search method used in our experiment runs in such a simple setting, allowed to access the global state in each step._
> >
> > Adding a sentence or footnote with this detail and comparing it with the information available to your approach will help the reader better understand why heuristic search is an upper-bound and not a reasonable solution by itself.
> >
> > From A7: _Although the ToM network of an agent has ToM models for each of the other agents, these models are actually shared because all the agents are homogeneous_
> >
> > I think this should be included in the paper because it is a very key detail.  This doesn't reduce the scalability demonstrated in the Observation Encoder and Message Sender networks (which are important demonstrations in your results), but it is important for the ToM network since Fig. 2 implies there are separate, unique models for each other agent.  Sharing parameters for each neighbor means the training is easier than learning separate models for each neighbor.  Since different neighbor agents are in different locations, I'm also curious why the neighbors can be considered to be homogeneous since they have unique areas of coverage in the tracking problem.  The goals of the top left agent are going to be different than the goals of the bottom right agent.

---

> > > ### Author Response · Authors · 2021-08-25
> > > **Reply to Reviewer ADoL**
> > >
> > > Thank you for your constructive suggestions. We will include the key details (additional results and clarifications) in the main paper in the revision.
> > >
> > > For the concern of baseline comparison, we implemented PPO in the multi-sensor multi-target tracking environment, and the coverage rate of PPO is **$66.58\pm0.45$**. It should be emphasized that PPO also benefits from the hierarchy structure. Note that our method ToM2C(optimized by A2C) reaches **$75.38\pm0.57$** (see Tab.2 in appendix), showing a prominent advantage over vanilla PPO. Besides, if PPO(instead of A2C) is implemented to optimize ToM2C, the coverage rate reaches **$75.09\pm0.85$**, close to the coverage rate of ToM2C(A2C). These results also show that ToM2C is compatible with different RL algorithms.
> > >
> > > For the concern of homogeneous agents, it would be better to refer to a formal definition in multi-agent system:
> > > - *All individual agents have the same goals, domain knowledge, and possible actions. They also have the same procedure for selecting among their actions. **The only differences among individual agents are their sensory inputs and the actual actions they take,** or in other words they are situated (placed) differently in the environment.*
> > >
> > > In our problems, all the agents are of the same action/goal space, observation space, and team reward. The observation of each is represented in the agent-centric coordinate. The difference in location and coverage area for agents is regarded as **different situations** in the environment. The agents are also of the same decision making procedure (neural network architecture). So they are homogeneous, sharing parameters among them is reasonable.

---

> ### Author Response · Authors · 2021-08-30
> **Note to Reviewer ADoL**
>
> In our response, we provided the details requested about the method, updated evaluation results and answered the questions. Please let us know if any of your concerns were not addressed by our response.

---

### Official Review · Reviewer_RvLm · 2021-07-27

**Rating:** 7
**Confidence:** 4

**Summary:**

* The paper proposes a Target-oriented Multi-agent Communication and Cooperation mechanism (ToM2C) using the Theory of Mind.

* It uses an observation encoder, ToM net, message sender, and decision-maker. The ToM net is designed for estimating the observation and inferring the goals (intentions). The message sender uses a graph neural network using inferred state and intention to generate local graphs. Once the agent receives the messages, it decides its goals using the inferred goals of others and received messages using the actor-critic.

* The empirical results of the model on the multi-sensor multi-target covering scenario is presented and compared with 4 different baselines from state-of-the-art MARL methods - HiT-MAC, I2C, A2C  and Heuristic Search algorithm.


**Limitations And Societal Impact:**

* The authors present the limitations of the research by describing that their model was evaluated in a simulated scenarios only which can be extended to real world environments for the robotic agents.
* It is acknowledged by the authors that their framework generated pseudo labels for communication reduction which can be optimized in future work.


**Main Review:**

Originality
* There is lot of research around Theory of Mind such as Machine Theory of Mind, multi-agent cooperation using theory of mind and ToM Net for two-player scenarios where an agent model another agent to infer its goal and preferences. The famework presented in this paper provides the modeling for multi-agent scenario consisting of multiple agents in a team where each agent is required to choose a subset of interesting targets and reaching them to contribute to the team goal.

* The communication reduction method used by the agents in the team that uses centralized training decentralized execution is interesting and makes the contributions novel. The authors train the connection choice network which provides the communication protocol for ’when’, ’who’ and ’what’ in multi-agent communication.


Quality
* The scenarios used in the multi-sensor multi-target tracking environment makes the environment much more realistic for a goal-oriented setting especially as agent can only obtain the information from its local observation.

* The training strategies 1) curriculum learning strategy to increase episode length L and γ factor during the training and 2) to split the optimization of the ToM and RL model to avoid the nested loop of influence among  the ToM net and the policy network is clearly defined and implemented in the code attached to the paper. The environment and model are implemented in Python.The model is built on PyTorch and was trained on a machine with 7 Nvidia GPUs (Titan Xp) and 72 Intel CPU Cores.

* The literature review is thorough and helps in understanding the current state of the art to find the missing gaps.


Clarity

Overall the paper is well organized and clearly written. The implementation details are clear. However, there are elements of the paper which do not provide enough information, such as:
* In order to obtain edge and compute the probabilistic distribution over the type of the  edge the authors propose to use the Interaction Networks (IN) by propagating the node and edge features spatially. However, the details about the marshalling function, m, that computes the matrix with interaction terms is missing? Additionally, which were the learnable parameters used in the objective function?

* To sample the discrete edge type in the local communication graph for the gradients to be back-propagated for training, the Gumbel-Softmax trick, is used in the message sender module. The authors need to provide as to what was the softmax temperature, tao, used in the model? It is needed because softmax temperature is a learned parameter as it can be interpreted as entropy regularization where the Gumbel-Softmax distribution can adaptively adjust the confidence of proposed samples during the training process.

* In the Inferred-goal Filter, the authors propose to generate local graphs using inferred state and intentions for agent i. The filtered feature is concatenated with the estimated observation representation to form the estimated node feature (equation 1) to form the estimated node feature. However the probability threshold for the inferred goal is not defined and needs to be explained.  Furthermore, for the agent i, in the observation estimation, how would the observation be made by i,  if the other agents are not facing(pose) towards the target?


Significance
* The submission provides the empirical results and compares them against the baselines from the existing MARL architectures:HiT-MAC, I2C, A2C  and Heuristic Search algorithm. They also provide the framework for modeling in multi-agent scenario consisting of n(>3) agents in a team where each agent is required to choose a subset of interesting targets and reaching them to contribute to the team goal.

* The pytorch implementation for the environment and the model is clear.

**Time Spent Reviewing:**

24

---

> ### Author Response · Authors · 2021-08-10
> **# Response to Reviewer RvLm**
>
> # Response to Reviewer RvLm
>
> Thank you for your detailed comments and helpful suggestions, we are encouraged that you value the novelty, clarity, and reproducibility. We address your specific questions and concerns in the comments below.
>
> - **Q1: The details about the marshaling function.**
> A1: All local graphs are complete directed graphs, so each node has a connection linked to every other node. Therefore, the marshalling function $m$ concatenates nodes features and edge features of each connection to form the interaction term $(V_s,V_r,E_{s,r})$, which is a $(2 D_v + D_E) * (N (N-1))$ matrix, where $D_v$ and $D_E$ are the length of node and edge feature vector, and $N$ is the number of agents (nodes).
>
> - **Q2: Which were the learnable parameters used in the objective function?**
> A2: As we mentioned in the submission, we adopted the training strategy that separates the learning of ToM net and other parts of the model. For ToM net, it is optimized according to the objective function in Eq. 6. The objective is composed of GI loss and OE loss, which are respectively related to the Goal Inference module and Observation Estimation module. The two modules are parameterized by $\theta_{GI}$ and $\theta_{OE}$, which are updated during ToM training. The other parts of the model, including observation encoder, message sender, and decision maker, are parameterized by $\theta{other}$ and updated with regard to RL loss (policy loss and value loss).
>
> - **Q3: Gumbel-softmax temperature.**
> A3: As[25], we did not make the softmax temperature $\tau$ learnable. It is constant and fixed to $0.5$ during the training and test process.
>
> - **Q4: The probability threshold for the inferred goal.**
> A4: The probability threshold $\delta$ is used to discretize the inferred probability to a binary choosing state (1: choose 0: not choose). For example, if agent $i$ infers that target $q$ has a probability of more than $\delta$ to be chosen by agent $j$ as its goal, the feature of the target $q$ will be added to the node feature of agent $j$ in the local graph $i$. Otherwise, the target feature will not be added to the node feature. To be specific, we set the $\delta$ to $0.5$ in this paper.
>
> - **Q5: How would the observation be made by $i$, if the other agents are not facing(pose) towards the target?**
> A5: All agents have access to the pose (camera direction in the tracking task) and location of each other agent. If a target is in the observation field of agent $i$, it will be able to infer whether this target is in the view of agent $j$ based on the location of the target as well as the location and pose of agent $j$. However, it is possible that some targets are out of the observation field of agent $i$, making it hard for agent $i$ to infer whether they can be observed by other agents. The Observation Estimation module is introduced to do the inference as correctly as possible, but the property of local observation inevitably leads to some errors in the inference.

---

> ### Author Response · Authors · 2021-08-30
> **Note to Reviewer RvLm**
>
> In our response, we provided the details requested about the method and answered the questions. Please let us know if any of your concerns were not addressed by our response.

---

> ### Comment · Reviewer_RvLm · 2021-08-31
> **Thank You for the clarification.**
>
> I appreciated the rebuttal discussion and the effort the authors put in. My main concern had to do with learnable parameters used in the objective function and clarification on the observation estimation of agent i. That has been addressed, so I am updating my score.

---

### Decision · Program_Chairs · 2021-09-27

**Decision:**

Reject

**Comment:**

Overall there is not enough support from reviews for me to recommend acceptance.

During the discussion, two reviewers bumped their respective scores by a point, but the overall scores remain in the borderline range.

Reviewers agreed that there were clear strengths to the paper, and that author responses cleared up some (but not all) issues. The main remaining concerns with the submission are
1. discussion of limitations, especially around the ToM modeled and how it compares to ideas in the literature,
2. providing experiment details/extensions discussed in the review responses, and
3. that making these revisions (and others suggested in the discussion) would constitute a major revision (and thus perhaps require another set of reviews).

On #1:
* Reviewer koJ6 wrote "the concepts of theory of mind [...] are treated somewhat casually and some examination (and citing) of the literature in this area would improve the paper".
* Reviewer 2f7u agreed "the ToM modeled by this work is quite rudimentary" and the fix would be "a good statement of limitation and a reframing of the contributions".
* Reviewer 2f7u also wrote "I would like to see more discussions on the limitations of the proposed method and how it is grounded in prior literature", and "the additional discussions on prior work and the experimental results provided in the authors' responses are valuable".

On #2, there were new results and many experiment details provided in the authors' rebuttal which should be added to the submission text, as reviewers suggested and agreed by the authors.

On #3, Reviewer 2f7u summarized "the additional experimental results are very important for people to really understand how this work is different from the prior work, what were the settings for the experiments, and how general this approach is (i.e., what kinds of problems could benefit from it). There should be a major revision to address these points, both in literature review and in experiments." I agree that including the content covered in the discussion period would require significant revisions to the paper.

Additionally:
* Reviewer ADoL reamined concerned about the treatment of scalability experiments, though these concerns were not highlighted by other reviewers.
* The main concerns of Reviewer RvLm, around learnable parameters and observation estimation, were addressed, and that score was raised to a 7 (the highest among the reviewers).

These issues must be weighed against the significance of the contributions in the paper, which reviewers found to be valuable but limited:
* Reviewer koJ6: "I'm not sure the results here are significantly better than heuristic search or some of the other baselines to conclusively state that the approach taken will make a significant impact on the field"
* Reviewer ADoL: "My greatest uncertainty after reading the paper is about the significance of its contributions to the literature. [...] Also, only one domain was considered, so it is difficult to assess how the benefits of the approach might generalize"
* Reviewer 2f7u: "The evaluation was done in only one experiment"


To summarize, reviewers agreed that the underlying work has strengths and that the paper is close to acceptance, but that to push the manuscript to be a clear acceptance would require substantial revisions.

Because substantial revisions are required, this submission doesn't quite meet the bar given the other strong submissions.